# Development of Proniosome Gel Formulation for CHIKV Infection

**DOI:** 10.3390/pharmaceutics16080994

**Published:** 2024-07-26

**Authors:** Ayça Altay Benetti, Ma Thinzar Thwin, Ahmad Suhaimi, Ryan Sia Tze Liang, Lisa Fong-Poh Ng, Fok-Moon Lum, Camillo Benetti

**Affiliations:** 1Department of Pharmacy and Pharmaceutical Sciences, National University of Singapore, Singapore 117544, Singapore; ayca.ben@nus.edu.sg (A.A.B.); e1268032@u.nus.edu (M.T.T.); e1123682@u.nus.edu (R.S.T.L.); 2A*STAR Infectious Diseases Labs (A*STAR ID Labs), Agency for Science, Technology and Research (A*STAR), Singapore 138648, Singapore; ahmad_suhaimi@idlabs.a-star.edu.sg (A.S.); lisa_ng@idlabs.a-star.edu.sg (L.F.-P.N.)

**Keywords:** osteoarthritis, CHIKV infection, berberine, proniosome, antioxidative, anti-inflammatory

## Abstract

Given the increasing aging population and the rising prevalence of musculoskeletal diseases due to obesity and injury, urgent research is needed to formulate new treatment alternatives, as current options remain inadequate. Viruses can exacerbate arthritis and worsen symptoms in patients with pre-existing osteoarthritis. Over the past decade, the chikungunya virus (CHIKV) has emerged as a significant public health concern, especially in Asia and South America. Exploring natural products, such as berberine, has shown promise due to its anticatabolic, antioxidative, and anti-inflammatory effects. However, berberine’s low stability and bioavailability limit its efficacy. We hypothesized that encapsulating berberine into a proniosome gel, known for its ease of preparation and stability, could enhance its bioavailability and efficacy when applied topically, potentially treating CHIKV infection. Our investigation focused on how varying berberine loads and selected excipients in the proniosome gel influenced its physical properties, stability, and skin permeability. We also examined the biological half-life of berberine in plasma upon topical administration in mice to assess the potential for controlled and sustained drug release. Additionally, we analyzed the antioxidant stress activity and cell viability of HaCaT keratinocytes and developed a lipopolysaccharide-stimulated cell culture model to evaluate anti-inflammatory effects using pro-inflammatory cytokines. Overall, the research aims to transform the treatment landscape for arthritis by leveraging berberine’s therapeutic potential.

## 1. Introduction

Musculoskeletal conditions refer to conditions that have an impact on joints, such as rheumatoid arthritis and osteoarthritis, bones, such as osteopenia and osteoporosis, and muscles, such as sarcopenia [1]. The significance of these musculoskeletal conditions is highlighted by a conducted analysis of Global Burden of Disease in 2019, whereby approximately 1.71 billion people are suffering globally due to restricted mobility and reduced levels of well-being. Furthermore, musculoskeletal conditions contributed immensely to years lived with disability (YLDs), as they account for 17% of around 149 million YLDs globally. This journal article primarily focused on osteoarthritis (OA), a degenerative joint disease, which progressively exacerbates and usually leads to chronic pain [2]. OA is the most prevalent form of arthritis, which impacts about 500 million people globally and, as of 2019, it ranked as the 15th dominating cause of YLDs [3]. This is a rising concern, as with the escalating rates of the aging population coupled with obesity and injury, the prevalence of OA will incessantly increase [4]. Besides factors such as aging, obesity, and injury fueling the prevalence of arthritis, it is also noteworthy that arthritis can be caused by viruses, such as chikungunya (CHIKV infection) [5], an arbovirus transmitted via *Aedes aegypti* mosquitoes, whereby almost all infected individuals suffer from momentary inflammation in affected joints, which can simulate more severe and tenacious forms of non-infectious arthritis [6,7]. Based on a review published in the journal *Clinical Medicine* in 2016, approximately 1% of acute arthritis cases are caused by viral infections [8]. With the quick onset of viral inflammation, patients diagnosed with pre-existing OA may suffer from an exacerbation of joint symptoms, such as stiffness and pain, which can continue to persist for years even after the clearance of viral infection [9]. The pathophysiological changes observed in OA encompass gradual degeneration of articular cartilage, subchondral sclerosis, osteophyte formation, and synovial inflammation [10]. OA remains incurable, as the current treatment landscape only offers temporary pain relief and inflammation reduction, and adverse events can further aggravate this chronic disease. The available treatment modalities for viral arthritis also focus on providing temporary relief of joint symptoms, through agents that similarly include non-steroidal anti-inflammatory drugs (NSAIDs) and analgesics [6,8]. However, corticosteroids are typically avoided due to their potential to mask or aggravate the underlying viral infection [11].

Of note, pharmacological evaluations have shown that the use of berberine outperforms mainstream pharmacotherapies in terms of diminishing inflammation and alleviating pain, while concurrently having a lower toxicity profile [12,13,14,15,16]. Nonetheless, a significant impediment in substantiating the therapeutic value of berberine is the inherent variability in treatment regimens, the lack of standardized formulations, and the imprecision of dosage administration. Among various chemical penetration enhancers, non-ionic surfactants (tweens, spans, etc.) are noteworthy for their capability to form niosome particles. Niosomes are structured from non-ionic surfactants that organize into vesicles, characterized by an aqueous core surrounded by a lipid bilayer (Figure 1). This aqueous core can encapsulate berberine, potentially offering a sustained-release mechanism. On the other hand, the non-ionic surfactants can enhance skin permeability by temporarily altering the lipid composition of the stratum corneum [17], which may facilitate increased berberine permeation. Additionally, niosomes were initially developed and patented by L’Oréal for cosmetic usages [18], demonstrating their effectiveness in dermal delivery. Nevertheless, niosomes may encounter physical stability issues due to the aggregation of non-charged surfactants.

In this research, proniosome, an anhydrous and free-flowing formulation of a carrier coated by water-soluble surfactants, was the chosen drug delivery system. Proniosomes can readily be reconstituted with an aqueous solvent prior administration or hydrated by skin to form niosomes (Figure 1), which can enhance the residence time of the drug in the stratum corneum and epidermis and promote skin permeation [17,19,20]. Proniosomes are relatively more stable and versatile than other vesicular systems, such as liposomes, due to less aggregation, while allowing comparable loading with hydrophilic and lipophilic drugs [19]. To leverage on and enhance the pharmacological effect of berberine despite its low stability and bioavailability, we explored how different amounts of berberine with selected excipients encapsulated into a proniosome gel affect the physical properties, stability, pharmacological effect, and skin permeability of this semi-solid dosage form. The rationale for selecting excipients, such as hyaluronic acid, ascorbic acid, resveratrol, and menthol, is highlighted in Table 1. We also determined the optimal loading dose of berberine to be evaluated in a CHIKV-infected mice model. Moreover, the biological half-life of berberine in plasma upon topical administration of the berberine proniosome gel in mice was investigated to determine whether our chosen drug delivery system could achieve a controlled and sustained drug release for CHIKV infection.

## 2. Materials and Methods

### 2.1. Chemicals and Materials

Berberine hydrochloride (molecular weight: 371.81 g/mol), Span^®^80, (1R, 2S, 5R)-(−)-menthol, and lipopolysaccharides from *Escherichia coli* were purchased from Sigma-Aldrich, Singapore. Tween^®^20 was purchased from Sinopharm Chemical Reagent Co., Ltd., Shanghai, China. L-ascorbic acid and cholesterol were purchased from Tokyo Chemical Industry Co., Ltd. (TCI), Tokyo, Japan. Resveratrol was purchased from Shaanxi Sciphar Natural Products Co., Ltd., Singapore. Hyaluronic acid (Mw: 25,000) powder was purchased through Sigma Aldrich, Singapore. Phosphate-buffered saline (PBS) was obtained from Vivantis, Seattle, WA, USA. The water employed in these experiments was purified using the Adrona Crystal EX water systems (Riga, Latvia). Dulbecco’s Modified Eagle Medium/Nutrient Mixture F-12 (DMEM/F-12), Dulbecco’s Modified Eagle Medium (DMEM), and Fetal Bovine Serum (FBS) were sourced from Hyclone, Thermo Fisher Scientific Inc. (Waltham, MA, USA).

### 2.2. Preparation of Berberine Proniosome Gel

The proniosome gel was formulated by adapting a previously documented method [19]. In summary, non-ionic surfactants (Span^®^80 and Tween^®^20) and cholesterol (10:1) were inserted in a glass vial. Subsequently, 125 µL of isopropanol (IPA) was added into 100 mg of a non-ionic surfactants:cholesterol mixture, and was heated at 65 °C for 5 min. Next, 80 µL of Milli-Q water was incorporated into the mixture, and it was again heated at 65 °C for 5 min. The samples were then removed, and the proniosome gel was allowed to cool to room temperature, facilitating the formation of the gel structure. Meanwhile additional excipients were added into the formulation, as listed in Table 1.

### 2.3. Rheological Characterization

Amplitude sweep and viscosity of berberine proniosome gel formulations were determined using a modular compact rheometer. The rheological characteristics of the berberine proniosome gel were determined using an Anton Paar Modular Compact Rheometer (NUS Centre for Additive Manufacturing, Singapore) with a cone-plane geometry of 50 mm diameter and at a 2° angle. Tests conducted on the berberine proniosome gel formulations included the amplitude sweep test and flow sweep test. The amplitude sweep experiment was performed at a fixed frequency of 1 Hz, with the strain amplitude ranging from 0.01% to 100%. The variations in the storage modulus (G′) and loss modulus (G″) across this range were assessed to identify the linear viscoelastic region (LVR). From this analysis, a strain amplitude of 0.03%, falling within the LVR, was selected for further frequency-dependent studies. The flow sweep test was performed between 0.1 s^−1^ and 100 s^−1^. Following a 1 min equilibration period at 25 °C, an oscillation frequency sweep test was executed. The oscillatory frequency varied from 0.1 rad/s to 100 rad/s, conducted at a constant strain of 0.03% within the linear viscoelasticity domain.

### 2.4. Preparation of Porcine Skin Tissues and Ex Vivo Skin Permeation Study

Ear tissues from pigs were donated by the School of Medicine under the NUS Animal Tissue Sharing Program. The specimens were thawed and cut into 2.5 cm × 2.5 cm squares using a D42 dermatome (Humeca Dermatome, Auxano Medical Pte Ltd., Singapore). The thickness of each specimen was measured to be 1.1 mm using a vernier caliper. The subcutaneous fats under the skin were removed followed by the trimming of hairs on the skin tissues using surgical scissors. Porcine skin was mounted on a vertical Franz diffusion cell with an exposure area of 1.21 cm^2^. The receptor compartment of the Franz diffusion cell was filled up with 5 mL of PBS and contained a magnetic stirrer bar to ensure continuous stirring of solution. The experiment was conducted at a temperature of 34 °C, and a stirring rate of 100 rpm with an infinite dose of loading. At time intervals of 0, 2, 14, 16, 18, 20, 22, 39, 44, and 48 h, an aliquot of 200 µL was withdrawn from the receptor compartment for sampling, followed by the addition of 200 µL of fresh PBS into the receptor compartment as a replacement.

### 2.5. UHPLC Analysis to Quantify Berberine in Skin Permeation Studies

From the aliquots obtained in Section 2.4. 50 µL was transferred into a HPLC vial and diluted with 150 µL of 50:50 PBS/acetonitrile. The diluted samples were analyzed by Nexera Shimadzu ultrahigh performance liquid chromatography (UHPLC) (Shimadzu, Kyoto, Japan). The method for UHPLC berberine quantitation was developed using a fluorescence detector. The excitation and emission wavelengths were 350 and 550 nm, respectively, as berberine was shown to exhibit weak intrinsic fluorescence with a maximum absorption around 530 nm within the range of 450 to 650 nm when excitation at 350 nm occurred. The samples were eluted through the Ascentis^®^ Express C18 column (7.5 cm × 4.6 mm, 2.7 µm) with a flow rate of 0.7 mL/min, using mobile phase: Milli-Q water and 0.1% formic acid (solvent A) and acetonitrile (solvent B) [31].

### 2.6. Cell Culture, Treatment, and Antioxidant Capacity Test Using HaCaT Cells

HaCaT keratinocyte cells were grown in Dulbecco’s Modified Eagle Medium (DMEM)/High Glucose supplemented with 10% Fetal Bovine Serum (FBS) in a T75 flask at 37 °C, under a humidified atmosphere of 5% carbon dioxide. The medium was changed on alternative days. Upon achieving confluency, HaCaT cells were harvested via trypsinization and seeded into a 12-well plate at a density of 1 × 10^5^ cells per well in 1 mL of medium. The 12-well plate was incubated for 24 h. After 24 h, 10 mg of each berberine proniosome gel formulation was reconstituted in 1 mL of PBS. The dissolved formulations were centrifuged at 4 °C, 300 G, for 5 min and sterilized with a 0.22 µm filter. HaCaT cells were treated with 30 µg/mL of berberine, and then incubated for 48 h [19]. To monitor the antioxidant stress activity and cell viability of HaCaT cells, the protocol from the ab65329 total antioxidant capacity assay kit (colorimetric) was performed accordingly [32].

### 2.7. LPS-Stimulated RAW264.7 Macrophage Cell Culture

To examine whether the formulations exhibited any anti-inflammatory effect, the LPS-stimulated RAW264.7 macrophage cell culture model was developed based on a modified protocol [33]. RAW264.7 macrophages (Figure 2) were seeded into a 6-well plate at a density of 1 × 10^5^ cells per well in 2 mL of medium and incubated for 24 h. These macrophages then induced inflammation with 25 ng/mL of LPS and were incubated for 22 h before treatment with 30 µg/mL of berberine, which was reported previously [19]. Dexamethasone was used as a negative control since it exhibits anti-inflammatory action by inhibiting TNF-α, IL-6, and IL-1β [34]. The cell viability assay was verified with the AlamarBlue assay protocol [35,36].

### 2.8. RNA Isolation and DNA Synthesis for Real-Time Quantitative PCR (rt-qPCR)

RNA isolation was performed using the TRI Reagent^®^ Protocol [37] and DNA synthesis was carried out using the Pure-NA First-Strand cDNA Synthesis kit [38]. PCR results were obtained via the CFX Connect^TM^ 7 Flex Real-Time PCR System (Bio-Rad, Singapore). A total of 500 ng of extracted mRNA was utilized to synthesize cDNA via the QuantiTect Reverse Transcription Kit [38]. Quantitative PCR was then conducted in triplicate on the CFX Connect™ 7 Flex Real-Time PCR System employing the iTaq Universal SYBR Green Supermix. The RNA expression levels of the samples were normalized to the reference gene glyceraldehyde 3-phosphate dehydrogenase (GAPDH). Subsequently, the relative gene expression of each gene for each formulation was calculated using the 2^−ΔΔCt^ method [36].

### 2.9. Ethics Approval

Three- to four-week-old gender-matched wildtype animals in C57BL/6J background were used in our in vivo CHIKV infection experiments. All experimental procedures were approved by the Institutional Animal Care and Use Committee (IACUC; #211635) of the Agency for Science, Technology, and Research (A*STAR), Singapore, in accordance with the guidelines of the Agri-Food and Veterinary Authority and the National Advisory Committee for Laboratory Animal Research of Singapore. All animals were purchased from Jackson Laboratory and were further housed and bred under specific-pathogen-free conditions in the Biological Resource Center of A*STAR.

### 2.10. Virus Stock

CHIKV (SGP011) was previously isolated from blood samples obtained from infected patients, admitted to the National University Hospital, Singapore, during the 2008 outbreak [39]. CHIKV was propagated in C6/36 cells (CRL-1660, ATCC©, Singapore), cultured in Leibovitz’s L-15 medium (Thermo Fisher Scientific) containing 10% Fetal Bovine Serum (FBS; Cytiva, Marlborough, MA, USA). CHIKV was subsequently concentrated and purified by ultra-centrifugation [40]. Titration of purified CHIKV was performed with VeroE6 (CRL-1586, ATCC©), cultured in Dulbecco’s Modified Eagle Medium (DMEM; Gibco, Waltham, MA, USA) containing 10% FBS (Cytiva) [40]. Purified CHIKV virus was aliquoted and kept at −80 °C for long-term storage. Both C6/36 and VeroE6 were tested for mycoplasma.

### 2.11. CHIKV Infection

Mice were inoculated subcutaneously with 1 × 10^6^ plaque-forming units (PFU) of CHIKV isolate diluted in 30 µL of PBS in the ventral side of the right footpad toward the ankle. Blood viral RNA load and joint-footpad swelling of the infected animals were monitored daily, as described previously [40,41,42,43]. Berberine proniosome gel (10 mg/mL) was applied thinly on the virus-inoculated joint-footpad starting from 1 day post-infection (1 dpi). Joint-footpad swelling is presented as the disease score reflecting the changes in footpad size post-CHIKV infection relative to baseline (non-infected). Blood viral RNA load was derived from 10 µL of tail blood diluted in 190 µL of PBS-citrate solution and extracted using the PrimeWay Viral RNA/DNA Extraction Kit (1st BASE). Viral RNA load was determined through the detection of the virus negative-strand nsP1 RNA [44]. qRT-PCR was performed using the Luna Universal Probe One-Step RT-qPCR Kit (New England Biolabs, Ipswich, MA, USA). The primers and probe sequences as well as thermal cycler conditions used have been previously reported [43]. Absolute viral RNA quantity was estimated from a standard curve generated using serial dilutions of synthetic CHIKV negative-strand nsP1 RNA transcripts, as reported [45]. Plasma was collected from sedated animals via orbital bleeding in Minicollect^®^ K3EDTA tubes (Greiner, Kremsmünster, Austria) at selected timepoints. Mice were sedated through inhalation of 1–3% isoflurane.

### 2.12. Plasma Berberine Quantification via LCMS

Chromatographic separation was performed using a C18 column (150 mm × 2.1 mm i.d., 5 μm) with water (A) and acetonitrile (B), both with 0.1% *v*/*v* of formic acid, as mobile phases at a flow rate of 0.35 mL/min. The gradient started at 85% of A for 1 min, switching at 90% of B at 4 min, and maintaining the same ratio until 6 min, going back to the initial conditions after that. The overall run time was 8 min, and the injection volume was 1 microliter. The column was maintained at 40 °C, while the autosampler was kept at 10 °C. The sample eluting from the Agilent 1290 Infinity II LC UHPLC system was introduced into an API 3500 triple-quadrupole mass spectrometer (Sciex, Framingham, MA, USA), operating in the positive mode with a needle voltage of 5.0 kV, using nitrogen as the nebulizing gas at 550 °C. Data acquisition and quantitative processing were accomplished with Analyst software, ver. 1.3.6. Optimal multiple reaction monitoring (MRM) conditions were 336–319 and 336–292, with a dwell time of 100 ms.

Berberine metabolites were analyzed in plasma samples after topical administration using the work from Qiu et al. [46], monitoring both Phase I and Phase II metabolites.

### 2.13. Statistical Analysis

For each experiment, three biological and three technical replicates were conducted. Obtained results were represented as mean ± standard deviation and statistical significance was analyzed using one-way ANOVA against a positive control via GraphPad Prism 10 software, whereby *p*-value ≤ 0.05 was regarded as statistically significant.

## 3. Results

### 3.1. Optimization of Preparation of Berberine Proniosome Gel

The berberine proniosome gel was prepared by modifying a previous protocol [19]. As our research aimed to develop berberine proniosome gel as a natural product for topical application, PBS was replaced with Milli-Q water. Table 2 displays the names of formulations prepared during the optimization process, which will be used for further discussion below.

### 3.2. Rheological Characterization

Amplitude sweep, frequency sweep, and flow sweep were conducted to investigate the impact on formulations when different amounts of berberine (0.5%, 1%, and 1.75%) were added with and without excipients. The shear-strain amplitude sweep was performed to describe the viscoelastic behavior of berberine proniosome gel. Shear strain refers to the deformation of the sample when shear stress, force per unit area, is applied [47]. This aimed to delineate the linear viscoelastic region (LVER) where samples could withstand the shear strain without being compromised, which is crucial for subsequent rheological evaluations. The amplitude sweep measurement is shown in Figure 3A. Accordingly, a shear strain of 0.03% was employed in further rheological characterizations since it resides within the LVER. From the amplitude sweep assessment, all formulations predominantly exhibited solid-state properties, indicated by higher G′ values compared to G″ within LVER. Extracting G’ values from the amplitude sweep measurement, these results validated that the incorporation of HA enhanced the malleability of the proniosome gel and increased its critical strain, while also notably increasing the G′ of 1.75% of berberine formulation (PG+B3) by approximately 10%. This increase can be attributed to the numerous hydroxyl, carboxyl, and N-acetyl functional groups along the HA polymer chain, which facilitate strong hydrogen bonding within the proniosome gel matrix. These interactions help to create a more cohesive network, improving the gel’s structural integrity and resistance to deformation [48,49,50]. With increasing amounts of berberine added, there was a decreasing trend in the exhibited viscoelastic solid behavior, as represented by lower yield points (Figure 3A).

With the incremental inclusion of berberine, a noticeable decrease in viscoelastic solid behavior was observed, as indicated by the reduction in yield points. This pattern became more evident with the incorporation of additional excipients, such as resveratrol, ascorbic acid, and menthol. The observed phenomenon could potentially be attributed to the oxidation processes involved. The explanation for this phenomenon using HA alone, without any excipients, implicates AA, which may induce structural deformation in the proniosome gel matrix [51]. Although, the oxidation and degradation mechanisms resulting from the combination of berberine with excipients, such as RES, AA, and MT, have not been elucidated in the literature. Meanwhile, viscosity refers to the resistance of fluid when undergoing deformation [52,53]. The 0.5% and 1% of berberine exhibited the highest viscosity, whereas PG+B2+AA+RES+MT exhibited the lowest viscosity (Figure 3B). This suggests that 0.5% and 1% of berberine are appropriate amounts to yield both a favorable gel-like texture and the greatest resistance to flow. Moreover, there was a delicate balance between berberine (solute) and solvent (water and IPA) interactions. At the optimal concentration, the interactions were maximized for viscosity enhancement. Beyond this point, the shift in balance may cause a reduction in viscosity due to less efficient solute–solvent interactions.

The frequency sweep results are often shown as in Figure 4, with storage modulus G′ and loss modulus G″ as the y-axis and angular frequency (rad/s) as the x-axis. Both axes are plotted on a logarithmic scale [54,55]. G’ represents the energy stored in a sample, whereas G″ represents the loss of deformation energy by internal friction during shearing. Additionally, when G′ > G″, the formulation is characterized as a viscoelastic solid with a gel-like structure, whereas when G″ > G′, the formulation is characterized as a viscoelastic liquid with a fluid-like consistency. This test was conducted to evaluate the spreadability of the formulations. The rheological characteristics of the proniosome gels were similar, exhibiting good elastic properties, with G″ increasing in accordance with the rising frequency. When this test was replicated with gels containing excipients (PG+B2+AA+RES+MT), the observed behavior across various berberine formulations remained relatively comparable, despite a noted decrease in both G′ and G″ values. This trend is consistent with results from previous amplitude and flow sweep analyses. A notable reduction in G′ was observed in the formulation with RES, AA, and MT, confirmed by amplitude sweep analysis, possibly attributable to the reaction between excipients. Moreover, at 0.1 rad/s, the gap between G′ and G″ became closer, and although G″ remained stable during the alteration of angular frequency, G′ slightly increased from 0.1 to 100 rad/s [56]. In stable proniosome gels, a network formed by intermolecular forces predominated, leading to a dominance of elastic behavior over viscous behavior, as evidenced by the nearly equidistant G′ and G″ values observed.

### 3.3. Ex Vivo Skin Permeation Study

To further investigate how formulations PG+B2+AA+RES+MT, PG+B3+HA, and PG+B3 impacted skin permeability, an ex vivo skin permeation study was conducted, as described in Section 2.4 and Section 2.5. It was concluded that the proniosome gel formulation significantly enhanced the skin permeability of berberine (Figure 5).

One-way ANOVA was performed to determine any significant difference in berberine permeated (%) after 48 h between PG+B2+AA+RES+MT, PG+B3+HA, and PG+B3. There was only a significant difference (*p* ≤ 0.033) between PG+B2+AA+RES+MT and PG+B3+HA (Figure 6A,B). This highlights HA as the best potential excipient to enhance the bioavailability of berberine compared to the combination of AA, RES, and MT. HA was added separately due to potential degradation by AA, as explained in Section 3.2 [51]. In addition, rheological and ex vivo studies revealed observable color changes and the degradation of the PG+B2+AA+RES+MT formulation. Consequently, long-term stability tests were performed in accordance with the ICH guideline Q1A(R2) [57]. The formulations were placed into glass containers for storage, maintaining a stable temperature of 25 °C and a relative humidity of 60%, facilitated by a climate chamber (Memmert ICH 110, Büchenbach, Germany). The long-term stability of the samples was evaluated at predefined time points: time zero (T0), 2 months (T60), 3 months (T90), and 6 months (T180). While the stability tests are still in progress to substantiate the concept of a stable proniosome gel, further analyses are scheduled to be conducted as part of the ongoing investigation.

Further analysis of cumulative permeation data indicated that the combination of these excipients did not significantly improve the delivery of berberine. The most promising results for berberine permeation were observed at a concentration of 1.75%, with no significant difference detected between formulations with and without HA. Generally, the impact of rheological properties on berberine permeation can be significant because they influence the formulation’s spreadability, adhesion to the skin, and berberine release characteristics. According to the permeation analysis of the proniosome gels, it was suggested that the rheological properties and viscoelastic behavior of the proniosome gel formulation do not notably affect the permeation profile of berberine.

Moreover, the stability of the formulations was tested under controlled conditions (25 ± 2 °C and 60% RH ± 5% RH). After six months, repeated skin permeation analyses demonstrated no significant difference in the amount of berberine permeated, attesting the stability of the formulation over time. These findings collectively highlight the potential for enhanced proniosome gel formulations in improving the transdermal delivery of berberine, providing a stable and effective therapeutic option [16,19,58]. Although the drug assay of berberine with the PG+B3 formulation indicated a significant reduction in berberine content (*p* ≤ 0.01), the ex vivo permeation profile remained consistent over time. Furthermore, the inclusion of HA into the formulation alone enhanced the stability of berberine within the proniosome gel matrix. This stability can be attributed to the viscoelastic properties and structural integrity of HA, which remain stable under prolonged storage conditions [59,60]. Nevertheless, the combination of RES, AA, and MT with berberine showed a significant reduction in the drug assay of berberine itself. This combination not only exhibited lower viscoelastic behavior compared to the other formulations but also reduced the amount of berberine over time and changed the color of the proniosome gel from yellow to gray.

### 3.4. Total Antioxidant Capacity and Cell Viability Assay

When assessing how different loadings of berberine impacted the antioxidant capacity and the cell viability of skin HaCaT cells, it was observed that there was a significant difference when the highest percentage (1.75%) of berberine was added (Figure 7A). Moreover, it was discovered that adding HA did not enhance the total antioxidant capacity based on PG+B3 and PG+B3+HA (Figure 7A). Hence, it could be concluded that most of the formulations did not have any acute toxicity and are safe for topical application on human skin cell keratinocytes (Figure 7B).

Considering that berberine is designed for dermal delivery, its cytotoxic profile on keratinocytes was also assessed in the previous studies [19]. As depicted in Appendix A, the concentration of berberine necessary to inhibit 50% of cell viability (IC50) was approximately 33 µg/mL via the MTT assay [61]. This finding is promising because the concentration of berberine needed to reduce NO levels is below the IC50 for keratinocytes. While 10 µg/mL of berberine is needed to significantly decrease the NO concentration, a lower concentration of berberine (1 µg/mL) suffices to restore sGAG production in chondrocytes, as illustrated in Appendix A. sGAG is a critical component secreted by chondrocytes to maintain cartilage tissue homeostasis. The in vitro results indicated that 1 µg/mL of berberine is adequate to induce a chondroprotective effect and restore sGAG production in the cells. Therefore, it can be concluded that the effective concentration range of berberine lies between 1 µg/mL and 33 µg/mL.

### 3.5. LPS-Stimulated RAW264.7 Macrophage Cell Culture with RT-qPCR

To investigate whether the formulations and chosen excipients (AA, RES, and HA) exhibited any anti-inflammatory effects, the change in levels of pro-inflammatory cytokines (TNF-α, IL-6, and IL-1β) was determined. It was evident that all formulations and excipients possessed anti-inflammatory properties, as they generally exhibited a significantly lower fold change than the positive control (Figure 8). However, formulations PG+B3 and PG+B2+AA+RES+MT and excipients RES and HA consistently displayed the highest statistical difference (*p*-value ≤ 0.0001) for TNF-α, IL-6, and IL-1β relative to the positive control.

The administered dose of berberine did not demonstrate a significant impact on the levels of TNF-α and IL-6. Importantly, there was no substantial variation in these markers across different berberine formulations. However, notable differences were observed, particularly with IL-1β, in response to varying doses of berberine. Specifically, formulations containing 1.75% berberine (PG+B3) and 1% berberine combined with excipients (PG+B2+AA+RES+HA) showed significant effects. These results can be attributed to berberine’s proven capacity to inhibit the expression of IL-1β [33]. HA can modulate the release of cytokines, which are signaling molecules that mediate and regulate immunity, inflammation, and hematopoiesis [24,62]. HA is known to selectively suppress pro-inflammatory cytokines TNF-α, IL-1β, and IL-6. This study supports that conclusion, although the anti-inflammatory effects of HA were not observed to be as significant as those of berberine when used alone in the formulation PG+B3. Notably, IL-1β demonstrated a marked change in its level of pro-inflammatory activity.

During the stability test phase, the formulation containing excipients PG+B2+AA+RES+HA demonstrated insufficient stability due to oxidation, as explained before. The three excipients exhibited differing reactivity and underwent oxidation over long-term storage, as shown in Figure 6C. Therefore, for further investigation in CHIKV-infected models, the PG+B3 formulation was selected for the in vivo testing on the mouse model.

### 3.6. CHIKV-Infected Pre-Clinical Model

Berberine, upon topical administration of the formulation of PG+B3 in mice, exhibited a biological half-life of 16 h (Figure 9). This is consistent with a slow permeation through the skin and the fact that the accumulation in the subdermal layer may adapt as a reservoir for controlled release, given that the half-life of berberine in plasma for intravenous injection in rats is short (0.22 h) [63].

In order to assess the efficacy of berberine proniosome gel as a therapeutic option for virus-induced arthritis, we turned to the CHIKV mouse model, a widely recognized system in arthritis research. Our investigation revealed promising results. Through daily application of the berberine proniosome gel at a concentration of 10 mg/mL, we observed a notable reduction in joint-footpad swelling, as illustrated in Figure 10a. Interestingly, despite this positive impact on inflammation, our analysis of viral RNA load, depicted in Figure 10b, did not exhibit significant differences. These findings underscore the potential of berberine proniosome gel in alleviating arthritic symptoms, positioning it as a promising avenue for further exploration in combating virus-induced arthritis.

### 3.7. Alanine Aminotransferase (ALT) and Creatinine Assay from Plasma Acquired from In Vivo Animal Study

To investigate whether the berberine proniosome gel exerts any toxic effects on the liver and affects kidney function, alanine aminotransferase (ALT) and creatinine levels in plasma acquired from the in vivo animal study were determined, respectively. Since the ALT levels in the control (healthy mice) typically ranged from 25 to 60 U/L, all groups of mice involved in the in vivo study exhibited normal plasma ALT levels (Figure 11) [63,64]. This suggests that the topical administration of berberine proniosome gel did not lead to any liver toxicity. Based on the comparison of creatinine levels between the control (healthy mice) and 0 h, whereby CHIKV infection was induced without topical administration of berberine proniosome gel, an initial increase in the creatinine levels was observed (Figure 11). However, after the topical administration of 1.75% berberine proniosome gel for 120 h, creatinine levels returned to baseline values, ranging from approximately 0.06 to 2.5 mg/dL. This observation indicates that the berberine proniosome gel exhibited reversible effects on creatinine levels, with statistically significant differences (*p* ≤ 0.05) observed between the control group and the 0 h time point, as well as between the control group and the 120-h time point.

The protective effect of berberine has potential implications for treating CHIKV and OA [16,19,33,65]. Berberine’s anti-inflammatory and antioxidant properties are well studied and likely contribute to its ability to modulate biochemical markers, such as creatinine. In the context of CHIKV, berberine may help reduce inflammation and oxidative stress within joint tissues and footpad swelling, thus alleviating symptoms and improving overall joint function. Similarly, these protective effects might extend to other pathologies characterized by inflammation and oxidative damage, potentially offering a non-invasive treatment for conditions ranging from chronic kidney disease to metabolic syndromes. Additionally, there are studies on the protective effect of berberine on creatinine levels and renal damage, especially in diabetic animal models [66,67]. This demonstrates that the proper formulation of berberine can enhance the delivery of this substance to the disease site, thereby increasing its protective effects against other pathologies. Further analysis will be conducted and the animal model for CHIKV will be adapted to serve as an OA disease model for comprehensive analysis. This dual approach will facilitate the investigation of both viral arthritis and OA through the administration of proniosome gel treatment.

### 3.8. Metabolism of Berberine Proniosome after Topical Gel Administration

Berberine metabolites were analyzed using the method described by Qiu et al. [46], but only jatrorrhizine (M3) and demethyleneberberine-2-*O*-β-d-glucuronide (M9) were detectable at 6 and 24 h after the administration. Quantitation was not feasible due to the lack of a reference standard. The specific detection of only jatrorrhizine (M3) and demethyleneberberine-2-*O*-β-d-glucuronide (M9) at 6 and 24 h post-administration could result from several factors: metabolic pathways, half-life, or slower elimination rates [68]. The administered dose of berberine and the resulting concentrations of its metabolites might lead to higher levels of M3 and M9 in the bloodstream. Further investigation will be conducted through in vivo animal studies to elucidate the underlying reasons for the specific detection of jatrorrhizine (M3) and demethyleneberberine-2-*O*-β-d-glucuronide (M9) at 6 and 24 h post-administration. These studies aim to provide a comprehensive understanding of berberine’s metabolic pathways, pharmacokinetics, and the influence of biological factors on metabolite formation and stability.

## 4. Discussion

### 4.1. Biological Mechanism of Berberine

Based on a study monitoring how berberine affects cartilage damage in IL-1β-stimulated rat chondrocytes and in a rat OA model, berberine was proven to inhibit expression of IL-1β and increase the expression of aggrecan and type II collagen, thereby promoting cartilage regeneration [15]. Additionally, a study utilizing articular chondrocytes and a post-traumatic OA model in mice showed that berberine not only restricts cartilage damage by lowering the expression of MMP-3 and MMP-13, but it also decreases the advancement of osteophyte formation and synovitis [65,69]. Berberine is also capable of inhibiting lipopolysaccharide (LPS)-induced inflammation in RAW264.7 macrophages through lowering the expression of TNF-α and suppressing the NF-kB signaling pathway via Sirtuin 1 (SIRT1)-dependent mechanisms, whereby SIRT1 deacetylates forkhead box O (FOXO) transcription factors responsible for maintaining detoxification of ROS [70,71]. Varghese et al. [72] consequently investigated the specific modifications induced by berberine on the cellular signaling environment that impede CHIKV replication. Their findings indicated that infection with CHIKV prompts the activation of all three principal MAPK signaling pathways—ERK, p38 MAPK, and JNK—in addition to the PI3K-Akt pathway. This study represents the first experimental evidence of concurrent activation of all major MAPK signaling pathways during CHIKV infection in human cells [16,72]. These studies indicate that berberine has promising protective efficacy against OA and CHIKV infection due to the demonstrated anticatabolic, antioxidant, antiviral, and anti-inflammatory effects. However, its pharmacological effect is limited by its low stability and absorption by the gastrointestinal wall, rapid metabolism, and clearance from blood plasma, thereby contributing to its low oral bioavailability, which was determined to be 0.36% in rats [58,73]. Moreover, the antiviral effect of berberine mostly depends on the dose of administration, and the intra-parenteral injection in mice showed a significant reduction in CHIKV infection and arthritis symptoms [16].

In this research, we aimed to identify suitable excipients that have the potential to enhance the treatment of viral arthritis and to study their physicochemical properties. We observed changes in rheological properties upon the addition of these excipients; however, no improvement in ex vivo permeation analysis was detected, irrespective of whether berberine was used alone or in combination with HA, RES, MT, or AA. This observation suggests that, despite the variations in the viscoelastic behaviors of our proniosome gel formulations, none demonstrated statistically significant permeation-enhancing properties. The most promising formulation was achieved with the incorporation of 1.75% berberine in the proniosome gel. Although the dose efficacy needs to be improved to better understand the mechanism of action of berberine in plasma, further studies are required to elucidate its metabolic pathways. This study presented different approaches to achieving a successful treatment of viral arthritis.

Besides that, the exploration of berberine proniosome gel as a therapeutic intervention for viral arthritis presents a compelling avenue for further investigation and optimization. While our initial findings demonstrated a moderate reduction in joint-footpad swelling without significant alterations in viral RNA load, there is ample opportunity to enhance the efficacy of this treatment regimen. One critical aspect to consider in optimizing the treatment regime is the dosage and frequency of application. Future research could focus on dose–response relationships to determine the optimal concentration for maximal therapeutic effect. Additionally, exploring alternative dosing schedules, such as varying the frequency of application or implementing a loading dose strategy, may provide insights into improving efficacy while minimizing potential adverse effects.

An interesting finding emerged from the ALT and creatinine analysis, revealing that the proniosome berberine formulation can mitigate the increase in creatinine levels induced by CHIKV infection. So far, the kidney protection effect of berberine has primarily been investigated in diabetic disease models [74,75]. This presents an interesting opportunity to explore its potential benefits in other conditions, such as viral arthritis. The initial findings of reduced creatinine levels in response to CHIKV infection, when treated with proniosome berberine gel, suggested that berberine may have broader renal protective effects beyond diabetes, warranting further investigation into its mechanisms and applications in musculoskeletal conditions.

Predominantly showing with musculoskeletal symptoms, CHIKV often leads to chronic polyarthritis, which can resemble autoimmune inflammatory arthritis [7,39]. Despite the availability of treatments, there is an unmet need for therapies that effectively reduce the viral load and alleviate joint inflammation. Notably, no specific antiviral therapy currently exists for CHIKV infection [5,16]. Furthermore, gaining a deeper understanding of the host–virus interactions in viral arthritis is essential for optimizing the use of berberine proniosome gel as a therapeutic agent. This includes elucidating the mechanisms by which viruses induce arthritis, the host immune response to viral infection, and how berberine may interfere with these processes. By unraveling the complex interplay between viral pathogenesis, host immune responses, and berberine treatment, we can identify key targets for intervention and refine treatment approaches to achieve better clinical outcomes.

### 4.2. Limitations of Current Applications of Berberine in Transdermal Delivery

Berberine has been utilized for centuries as a holistic approach to healthcare and healing via Traditional Chinese Medicine (TCM) [76]. However, there are some limitations to the current applications of berberine formulations that need to be addressed. The major limitation lies in the lack of scientific studies and evidence supporting the efficacy and safety of these formulations. Many Traditional Chinese Medicine formulations with berberine have not undergone rigorous clinical trials or met the rigorous standards of modern medicine. This lack of scientific validation impedes the wider acceptance of berberine in mainstream healthcare systems. TCM formulations often lack precise dosage recommendations; therefore, the absence of standardized dosage guidelines can result in improper usage and potential harm to patients. Additionally, the complex nature of berberine formulations poses challenges in understanding their individual mechanisms and potential drug interactions. This limitation increases the risks associated with polypharmacy and adverse reactions [12,77,78].

Nevertheless, the penetration of these new delivery systems into the commercial sphere and their accessibility to patients encounter numerous barriers. The main issue in berberine formulations lies in achieving consistency and stability, followed by the greater challenge of scalability at an industrial level within a Good Manufacturing Practice (GMP) setting [15]. Besides reproducibility and shelf-life, scaling up represents one of the most significant challenges in the commercial production of TCM products. Development and optimization at the laboratory scale might be comparatively straightforward but transitioning to large-scale production encounters numerous difficulties. Compared to traditional dosage forms that are well established, vesicular and particulate delivery systems exhibit a lower likelihood of successful technology transfer from laboratory to pilot production and then to large-scale manufacturing [79,80].

This work proved the efficacy of berberine-based proniosomes in addressing CHIKV infection thanks to the increased permeation due to the proniosomes and the stability over time of the final dosage forms. Those results can be translated to different drugs, based on existing TCM products, to make them suitable in modern medicine.

## 5. Conclusions

Our proniosome gel formulation, which can encapsulate both berberine and additional excipients to enhance the formulation’s efficacy, proved to be effective against CHIKV infection and stable over time, addressing some of the issues with TCM products based on natural products. We examined the impact of the amount of berberine, a natural alkaloid with proven pharmacological activities, and additional excipients encapsulated into the proniosome gel, to evaluate their skin permeability, their efficacy on CHIKV-infected preclinical animal models (as anti-inflammatory, thanks to its effect on cytokines, such as TNF-α, IL-6, and IL-1β, and antioxidant), and their stability. The proniosome gel was prepared using various excipients, including hyaluronic acid, ascorbic acid, resveratrol, and menthol. Each formulation was systematically evaluated to determine its physical properties, including viscoelastic behavior and viscosity. Hyaluronic acid emerged as significantly improving the formulation’s viscoelastic behavior and stability, while berberine displayed a strong antioxidant effect, which was a significant finding considering its minimal oral availability when administered topically. Moreover, hyaluronic acid significantly improved the formulation’s stability and anti-inflammatory properties, making it a crucial component of the proniosome berberine gel.

The slow permeation of berberine and its accumulation in the subdermal layer indicate a controlled-release mechanism. This is particularly beneficial for conditions such as viral arthritis CHIKV pathogenesis, where sustained drug release can enhance therapeutic outcomes of berberine as a natural compound. The 1.75% berberine proniosome gel proved to be the most promising formulation, exhibiting controlled and sustained drug release. An in vivo preclinical study using mice showed that the half-life of berberine upon topical administration is short, consistent with the data available in the literature. This suggested that the subdermal layer could act as a reservoir, enabling a controlled and sustained release of berberine. However, further studies are necessary to understand the mechanism of action of berberine in plasma and elucidate its metabolic pathway and protective effect. Continued research is essential to further optimize the proniosome gel formulation. Further studies should focus on exploring different concentrations of berberine and other potential excipients to enhance the gel’s efficacy.

## 6. Patents

There is a patent resulting from the work reported in this manuscript.

## Figures and Tables

**Figure 1 pharmaceutics-16-00994-f001:**
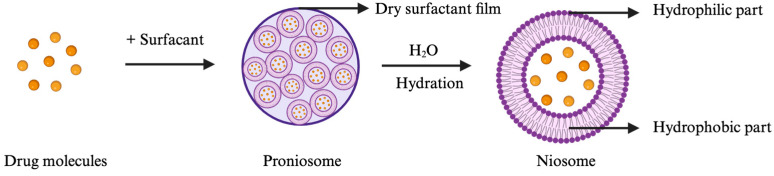
Formation of niosomes upon hydration of proniosomes with water (illustration created with BioRender.com).

**Figure 2 pharmaceutics-16-00994-f002:**
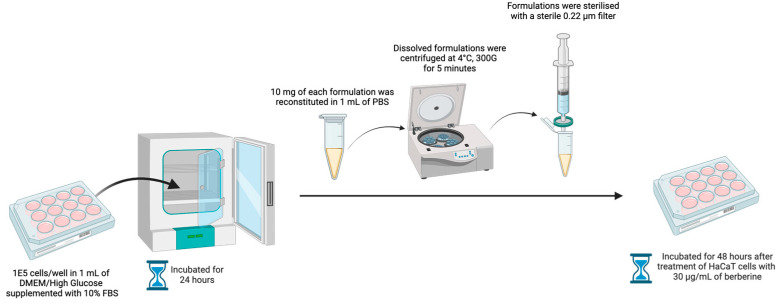
Illustration of the LPS-stimulated RAW264.7 macrophage cell culture model (illustration created with BioRender.com).

**Figure 3 pharmaceutics-16-00994-f003:**
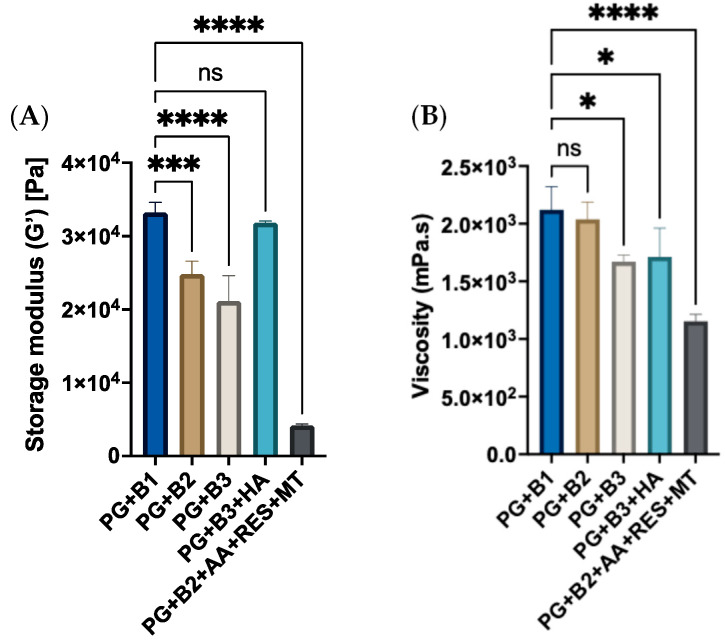
(**A**) Graph of storage modulus (Pa) of different berberine proniosome formulations with a shear strain of 0.03% (n = 3). (**B**) Viscosity (mPa.s) of different berberine proniosome formulations with a shear rate of 50 1/s (n = 3). PG+B1, PG+B2, and PG+B3: 0.5%, 1%, and 1.75% of berberine; PG+B2+AA+RES+MT: 1% of berberine, 1% of ascorbic acid, 1% of resveratrol, and 2% of menthol *w*/*w* for proniosome gel. PG+B3+HA: 1.75% of berberine and 1.75% of hyaluronic acid *w*/*w* for proniosome gel. One way ANOVA; *p* > 0.05 (ns: not significant), *p* ≤ 0.05 (*), *p* ≤ 0.001 (***) and *p* ≤ 0.0001 (****).

**Figure 4 pharmaceutics-16-00994-f004:**
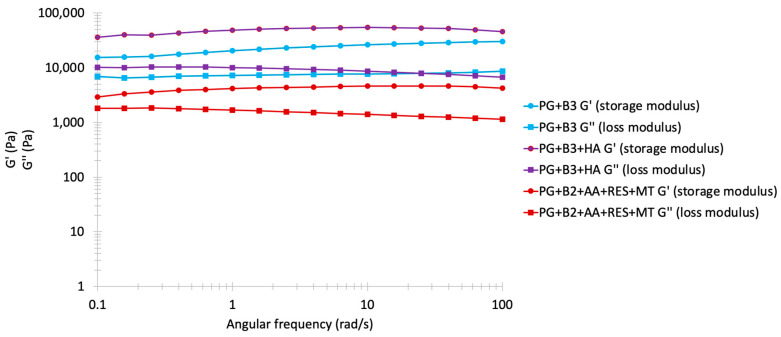
Frequency sweep analysis: storage and loss modulus (Pa) of PG+B3, PG+B3+HA, and PG+B2+AA+RES+MT (1% and 1.75% of berberine) proniosome formulations against angular frequency (rad/s; n = 3). PG+B2+AA+RES+MT: 1% of berberine, 1% of ascorbic acid, 1% of resveratrol, and 2% of menthol *w*/*w* for proniosome gel. PG+B3+HA: 1.75% of berberine and 1.75% of hyaluronic acid *w*/*w* for proniosome gel.

**Figure 5 pharmaceutics-16-00994-f005:**
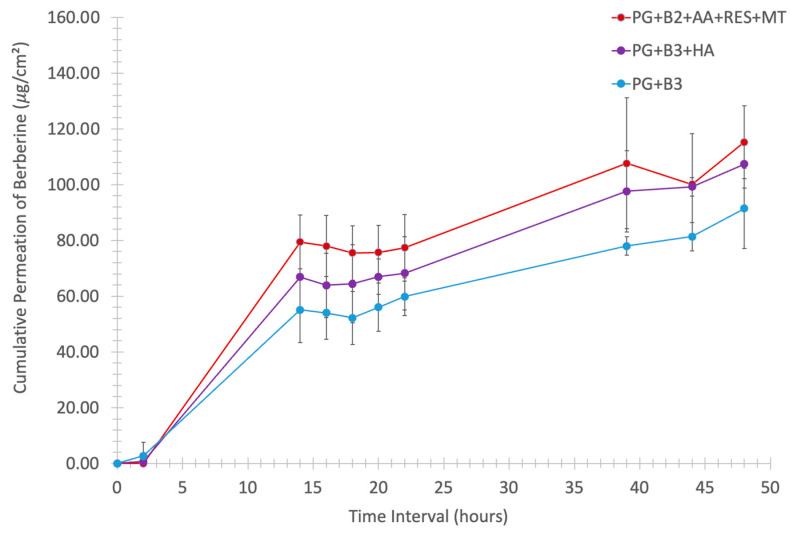
Cumulative permeation of berberine (μg/cm^2^) through porcine skin of thickness 1.2 mm in vertical Franz diffusion cells with an exposure area of 1.21 cm^2^ against the time interval (0, 2, 14, 16, 18, 20, 22, 39, 44, and 48 h) at 34 °C, 100 rpm (n = 6). PG+B2+AA+RES+MT: 1% of berberine, 1% of ascorbic acid, 1% of resveratrol, and 2% of menthol *w*/*w* for proniosome gel. PG+B3+HA: 1.75% of berberine and 1.75% of hyaluronic acid *w*/*w* for proniosome gel. PG+B3: 1.75% of berberine *w*/*w* for proniosome gel.

**Figure 6 pharmaceutics-16-00994-f006:**
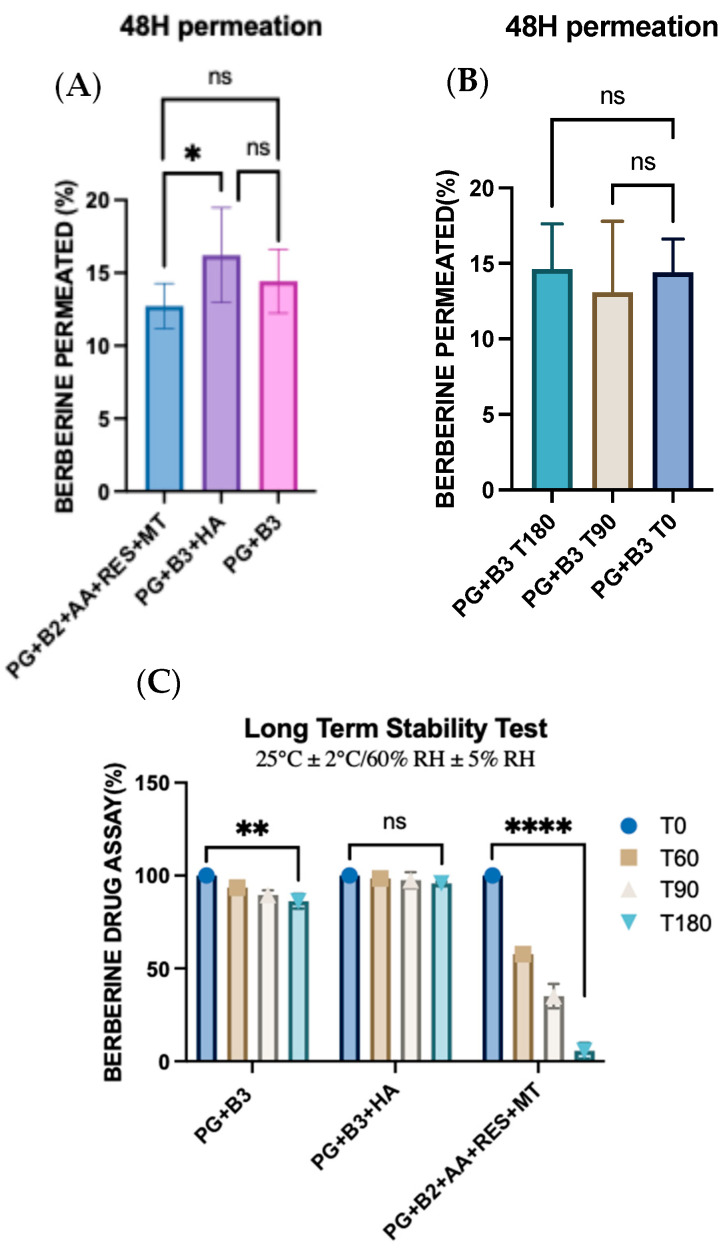
(**A**) One-way ANOVA test for berberine permeated (%) after 48 h between PG+B2+AA+RES+MT, PG+B3+HA, and PG+B3 (n = 3). (**B**) One-way ANOVA test for berberine permeated (%) after 48 h between stability of PG+B3 (time zero, after 3 months at room temperature, and after 6 months at room temperature; n = 3). (**C**) Long-term stability tests of berberine formulations at 25 °C and 60% RH over six months (T0: time zero, T60: 2 months, T90: 3 months, and T180: 6 months; n = 3). PG+B2+AA+RES+MT: 1% of berberine, 1% of ascorbic acid, 1% of resveratrol, and 2% of menthol *w*/*w* for proniosome gel. PG+B3+HA: 1.75% of berberine and 1.75% of hyaluronic acid *w*/*w* for proniosome gel. PG+B3: 1.75% of berberine *w*/*w* for proniosome gel. One-way ANOVA; *p* > 0.05 (ns: not significant), *p* ≤ 0.05 (*), *p* ≤ 0.01 (**), and *p* ≤ 0.0001 (****).

**Figure 7 pharmaceutics-16-00994-f007:**
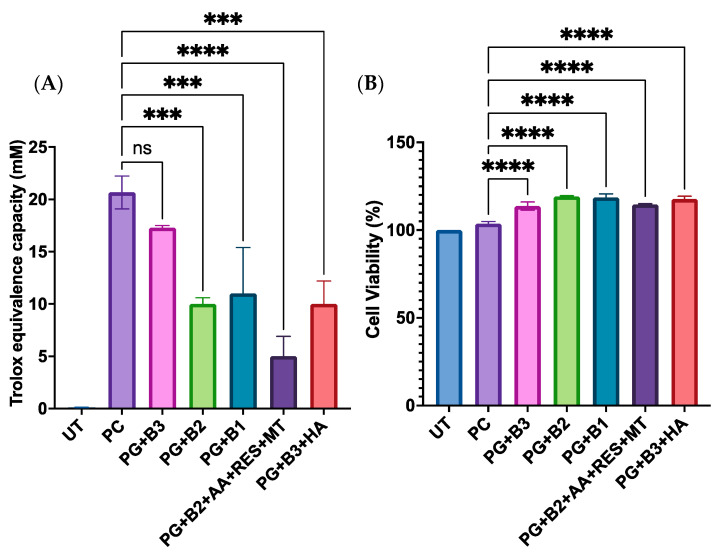
(**A**) Trolox equivalent capacity (mM) (3 technical replicates), whereby untreated control, positive control (Trolox), and proniosome gel formulations were compared to the standard, Trolox, according to the ab65329 total antioxidant capacity assay kit (colorimetric) protocol [32]. (**B**) Cell viability (%) of HaCaT cells after 72 h of incubation (n = 3). One-way ANOVA; *p* > 0.05 (ns: not significant), *p* ≤ 0.001 (***), and *p* ≤ 0.0001 (****). PG+B2+AA+RES+MT: 1% of berberine, 1% of ascorbic acid, 1% of resveratrol, and 2% of menthol *w*/*w* for proniosome gel. PG+B3+HA: 1.75% of berberine and 1.75% of hyaluronic acid *w*/*w* for proniosome gel. PG+B3: 1.75% of berberine *w*/*w* for proniosome gel. PG+B2: 1% of berberine *w*/*w* for proniosome gel. PG+B1: 0.5% of berberine *w*/*w* for proniosome gel.

**Figure 8 pharmaceutics-16-00994-f008:**
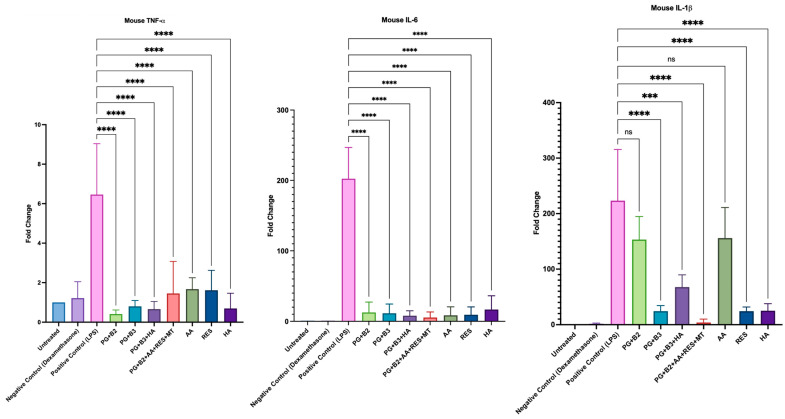
Plotted graphs of fold change against the respective mouse pro-inflammatory cytokine: TNF-α, IL-6, and IL-1β for positive control (LPS), negative control (dexamethasone), untreated cells, proniosome gel formulations—PG+B2+AA+RES+MT: 1% of berberine, 1% of ascorbic acid, 1% of resveratrol, and 2% of menthol *w*/*w* for proniosome gel; PG+B3+HA: 1.75% of berberine and 1.75% of hyaluronic acid *w*/*w* for proniosome gel; PG+B3: 1.75% of berberine *w*/*w* for proniosome gel; PG+B2: 1% of berberine *w*/*w* for proniosome gel; PG+B1: 0.5% of berberine *w*/*w* for proniosome gel—and excipients: AA, RES, and HA, respectively (n = 3). One-way ANOVA; *p* > 0.05 (ns: not significant, *p* ≤ 0.001 (***), and *p* ≤ 0.0001 (****).

**Figure 9 pharmaceutics-16-00994-f009:**
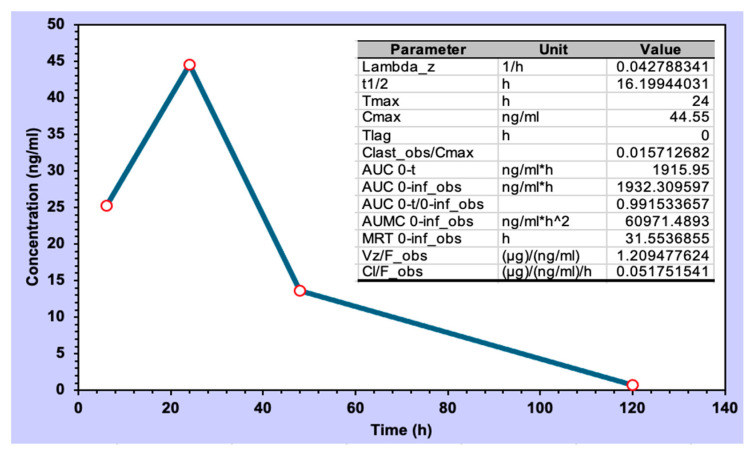
CHIKV infection pre-clinical model and pharmacokinetic analysis of berberine CHIKV infection pre-clinical model and pharmacokinetic analysis of berberine concentration (ng/mL). Sampling points were 6, 24, 48, and 120 h (n = 3; the sampling was collected from different infected animals at each time interval).

**Figure 10 pharmaceutics-16-00994-f010:**
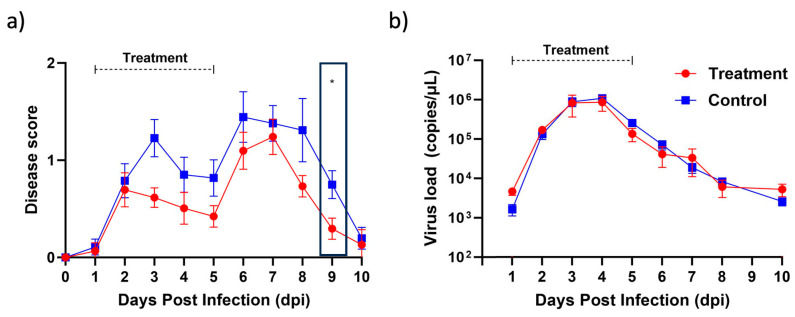
CHIKV infection pre-clinical model. Wildtype animals were inoculated with 1 × 10^6^ PFU of CHIKV in the right footpad. Between 1 and 5 dpi, 1.75% of berberine proniosome gel (10 mg/mL) PG+B3 was applied daily on the virus-inoculated footpad. (**a**) Joint-footpad swelling (n = 9 per group) was monitored daily over 10 days. (**b**) Viral RNA load in tail blood of CHIKV-infected animals (n = 9 per group) over a period of 10 days. One-way ANOVA; *p* > 0.05 (ns: not significant) and *p* ≤ 0.05 (*).

**Figure 11 pharmaceutics-16-00994-f011:**
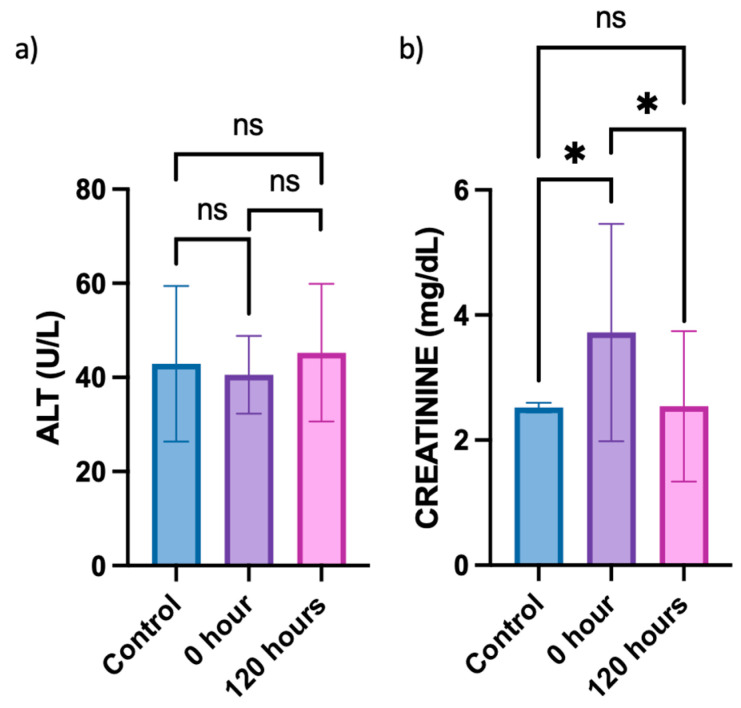
CHIKV infection pre-clinical model. (**a**) ALT plasma level of control mice vs. mice treated with 1.75% of berberine proniosome gel PG+B3 upon viral arthritis induction after 0 and 120 h. (**b**) Creatinine plasma level of control mice vs. mice treated with berberine proniosome gel upon viral arthritis induction after 0 and 120 h (n = 3 per group). One-way ANOVA; *p* > 0.05 (ns: not significant) and *p* ≤ 0.05 (*).

**Table 1 pharmaceutics-16-00994-t001:** Rationale for selecting hyaluronic acid, ascorbic acid, resveratrol, and menthol as excipients.

Excipient	Pathogenesis-Targeted Mechanism	Pharmacological Effect	References
Hyaluronic acid	Anticatabolic and anti-inflammatory	OA onset causes a lower concentration of hyaluronic acid in synovial fluid compared to healthy joints. Application of exogenous hyaluronic acid can promote chondrocyte synthesis of endogenous hyaluronic acid and proteoglycans, as well as decrease the production of pro-inflammatory mediators and MMPs. Hyaluronic acid can potentially limit cartilage damage by promoting its regeneration. Hyaluronic acid can be used to lubricate joints, exerting a protective effect, and is of then included as a component in infiltration protocols.	[21,22,23,24]
Ascorbic acid	Antioxidative	Ascorbic acid can prevent the differentiation of chondrocytes during oxidative stress. Ascorbic acid can limit cartilage damage by increasing the expression of proteoglycans and collagens.	[22,25]
Resveratrol	Anti-inflammatory and antioxidative	Resveratrol can inhibit the expression of pro-inflammatory cytokines: TNF-α, IL-1β, and IL-6, which decreases the production of iNOS, leading to lower NO levels. Resveratrol is an activator of SIRT1, thereby upregulating FOXO transcriptional activity to protect chondrocytes from oxidative stress. Resveratrol can limit cartilage damage by increasing the expression of proteoglycans and collagens.	[22,26,27,28]
Menthol	Pain	Menthol can serve as an analgesic agent to alleviate pain, as well as improving the smell of berberine proniosome gel.	[29,30]

**Table 2 pharmaceutics-16-00994-t002:** Berberine proniosome gel formulations with respective amounts of berberine and specific excipients (HA = hyaluronic acid; AA = ascorbic acid; RES = resveratrol; MT = menthol) added during preparation.

Berberine Proniosome Gel (PG) Formulation	Amount of Berberine (% (*w*/*w*))	Additional Excipients
PG+B1	0.5%	-
PG+B2	1%	-
PG+B3	1.75%	-
PG+B3+HA	1.75%	Hyaluronic acid (HA) (1.75%)
PG+B2+AA+RES+MT	1%	Ascorbic acid (AA) (1%) Resveratrol (RES) (1%) Menthol (MT) (2%)

## Data Availability

The datasets presented in this article are not readily available because the data are part of an ongoing study and a current patent application, which is granted.

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
