# Peer review of "Development of Proniosome Gel Formulation for CHIKV Infection"

_pharmaceutics, 2024, doi:10.3390/pharmaceutics16080994_

Round 1

Reviewer 1 Report

Comments and Suggestions for Authors

The manuscript deal with the formulation of a drug for the treatment of chikungunya virus. The study is composed of a deep physico-chemical evaluation as well as in vitro and in vivo evaluation. Introduction is well written and interesting. The problematic of viruses induced or exacerbated OA is well presented, as well as the rationale of the study. Experiments are logical and support the use of berberine formulations.

However, the text is not always logically written or easy to read, especially for the physiochemical part. I recommend a revision with a proofreading and make these suggestions:

- Introduction : the introduction is quite long, while it does not really present the proniosome. I would recommend shortening a bit the introduction and expand a bit the proniosome part.

-line 127: please precise the type of non-ionic surfactant (Span80?)

- line 128: “IPA” should be explained (isopropanol?)

- Table 1: Hyaluronic acid is also classicaly used to lubricate joints (protective effect), as a part of infiltration protocols. While it is not truly evidenced, it could be added as a potential effect.

-Table 2: - please harmonize the denomination for HA, RES, AA and MT (both acronyms and full names are used)

- What is the rationale of these formulations? For instance, is there any reason for not formulating PG+B2+HA? This could be stated.

- Line 285: you mention that you use the intersection of G’ and G” as the limit of the viscoelastic region. Usually, we consider the linear viscoelastic region (LVER) as the region where G’ is stable and does not decrease (network cleavage) upon deformation. The intersection is not really the LVER, but more a transition from a mainly elastic to a mainly plastic deformation, so please remove this statement. Also, moduli are dependent on the frequency: you choose to do the amplitude sweep tests at 10 rad/s, please justify this choice.

- The rheological evaluation: all of this part seems quite extensive while it is not much discussed in the text.

- What the expected results? Why the addition of excipients is decreasing the viscosity?

- Are the differences significative? Are there any errors?

- I recommend simplifying this part but better interpret.  For example, you could merge figures 3 and 5 and merge the figures 4 and 6. You could also only represent G’ (elasticity = strength of the gel) and/or tanDelta which is the ratio between the moduli (elasticity versus plasticity = gel state, but this is dependent on the frequency).

-  You work with proniosome gel. It seems that proniosome is a solid form. Please clarify the form of your system. Did you check the morphology of the niosome after hydration (DLS?)? How can you prove you don’t just have a monophasic hydrogel formulation?

Section 3.4: - It seems that you conclude that the antioxidant activity validates the absence of toxicity. Please rewrite.

                        - I think it is quite classic to perform a logarithmic dose/dependence evaluation of toxicity. Please perform this test or justify the dose you used. Same for the in vivo evaluation, why did you choose this dose?  

- Fig 10 is very small to read.

- Line 324 : you justify the addition of HA alone because it can be degraded by AA. Please specify the mechanism. This should also be justified earlier in the text.

- Line 447 : you mention stability tests to discard PG + B2 + AA + RES +HA. Where are these tests? Did do you determine a limit time for storage? This does refer to any mention elsewhere.

- Fig 12 : you mention a diminution of the footpad swelling. What could this difference mean for a patient? Would it be highly alleviating if transposed to OA ?

- Line 632 : in this part, you discuss ALT and creatinine levels. You make the link with between diabetes, kidney and arthritis… I don’t understand, this should be rewritten to clearly explain how the protective effect of berberine can be linked to viral arthritis or other pathologies.

- section 4.2 : this section, and especially the second paragraph, seem very long owing the goal of the study.

-Conclusion :

            - The conclusion starts by dealing with TCM. As TCMs are not part of the introduction, it seems the conclusion is disconnected from the introduction. Harmonization, clear objectives of the study (in the introduction) and a clear conclusion has to be drawn.

Author Response

Reviewer: - Introduction : the introduction is quite long, while it does not really present the proniosome. I would recommend shortening a bit the introduction and expand a bit the proniosome part.

Comment: The recommended corrections had been followed.

Reviewer: -line 127: please precise the type of non-ionic surfactant (Span80?)

Comment: Corrected

Reviewer: - line 128: “IPA” should be explained (isopropanol?)

Comment: Corrected

Reviewer: - Table 1: Hyaluronic acid is also classicaly used to lubricate joints (protective effect), as a part of infiltration protocols. While it is not truly evidenced, it could be added as a potential effect.

Comment: Corrected and added

Reviewer: -Table 2: - please harmonize the denomination for HA, RES, AA and MT (both acronyms and full names are used)

- What is the rationale of these formulations? For instance, is there any reason for not formulating PG+B2+HA? This could be stated.

Comment: The denomination was corrected and harmonized. Some of PG formulations have already been published in our previous studies. This PG formulation is protected thanks to a patent. The rationale behind adding excipients is to differentiate the formulation and to conduct research and development for possible combinations. We didn’t choose PG+B2+HA because, during this study, we realized that we could load more berberine and excipients, so we chose to maximize the loading of berberine to invastigate the efficacy without changing the physicochemical properties. Therefore we choose PG+B3+HA and PG+B2+HA+RES+AA+MT.

Reviewer: - Line 285: you mention that you use the intersection of G’ and G” as the limit of the viscoelastic region. Usually, we consider the linear viscoelastic region (LVER) as the region where G’ is stable and does not decrease (network cleavage) upon deformation. The intersection is not really the LVER, but more a transition from a mainly elastic to a mainly plastic deformation, so please remove this statement. Also, moduli are dependent on the frequency: you choose to do the amplitude sweep tests at 10 rad/s, please justify this choice.

Comment: Thank you for pointing that out. We confirm that the LVER region (ranging from 0.01% to 0.1% shear strain) with the PG+B1 formulation is accurately described. As noted, the incremental amounts of berberine and excipients make this region a transition from elastic to plastic deformation. This clarification has been added to the manuscript. There were some errors in describing the method of the amplitude sweep, which was incorrectly confused with the oscillatory frequency sweep. We did not include frequency test results. The necessary corrections have been made accordingly.Frequency sweep: After a 60-second equilibration period, the samples were subjected to oscillatory frequency sweeps ranging from 0.1 rad/s to 100 rad/s at a constant strain of 0.1%, within the linear viscoelastic region. The frequency range, along with the values of the storage modulus (G′) and the loss modulus (G′′), were plotted on a logarithmic scale.

Reviewer- The rheological evaluation: all of this part seems quite extensive while it is not much discussed in the text.What the expected results? Why the addition of excipients is decreasing the viscosity?Are the differences significative? Are there any errors?I recommend simplifying this part but better interpret.  For example, you could merge figures 3 and 5 and merge the figures 4 and 6. You could also only represent G’ (elasticity = strength of the gel) and/or tanDelta which is the ratio between the moduli (elasticity versus plasticity = gel state, but this is dependent on the frequency). You work with proniosome gel. It seems that proniosome is a solid form. Please clarify the form of your system. Did you check the morphology of the niosome after hydration (DLS?)? How can you prove you don’t just have a monophasic hydrogel formulation?

Comment: Thank you so much for the recommendations. There was no change in viscosity when the berberine was added to the formulation. The significant differences were demonstrated in the graphs for the amended changes. Although the decrease in viscosity for PG+B2+HA+RES+AA+MT can be explained by the degradation and oxidation of the final product. The proof of concept for our proniosome gel was performed, and the stability of the gels was tested. The results are added in the annex file.

Reviewer: Section 3.4: - It seems that you conclude that the antioxidant activity validates the absence of toxicity. Please rewrite. I think it is quite classic to perform a logarithmic dose/dependence evaluation of toxicity. Please perform this test or justify the dose you used. Same for the in vivo evaluation, why did you choose this dose?  

Comment: This study was reported in previous research as a PhD thesis. A supplementary figure is provided to explain the minimum toxic dose. 

Reviewer: - Fig 10 is very small to read.

Comment: The figure size was increased, original image will be given for final editing.

Reviewer: - Line 324 : you justify the addition of HA alone because it can be degraded by AA. Please specify the mechanism. This should also be justified earlier in the text.

Comment: The justification added in the line 295

Reviewer: - Line 447 : you mention stability tests to discard PG + B2 + AA + RES +HA. Where are these tests? Did do you determine a limit time for storage? This does refer to any mention elsewhere.

Comment: Stability testing was added in the Figure 6c.

Reviewer: - Fig 12 : you mention a diminution of the footpad swelling. What could this difference mean for a patient? Would it be highly alleviating if transposed to OA ?

Comment: Footpad swelling is a symptom of irritation, along with rubor, which are also symptoms of OA due to cartilage deficiency that overworks the joint and muscles. The animal model for Chikungunya virus (CHIKV) will be modified to serve as an OA disease model for further analysis. This way, viral arthritis and OA will be investigated using proniosome gel treatment. 

Reviewer: - Line 632 : in this part, you discuss ALT and creatinine levels. You make the link with between diabetes, kidney and arthritis… I don’t understand, this should be rewritten to clearly explain how the protective effect of berberine can be linked to viral arthritis or other pathologies.

Comment: We amended the paragraph to make it more consistent.

Reviewer: - section 4.2 : this section, and especially the second paragraph, seem very long owing the goal of the study.

Reviewer: The section was refined and the paragraph was shorted.

Reviewer: -Conclusion - The conclusion starts by dealing with TCM. As TCMs are not part of the introduction, it seems the conclusion is disconnected from the introduction. Harmonization, clear objectives of the study (in the introduction) and a clear conclusion has to be drawn.

Reviewer: We amended the paragraph to make it more consistent.

Reviewer 2 Report

Comments and Suggestions for Authors

The authors describe the development of a proniosomal gel for berberine HCl for topical treatment of CHIKV viral conditions. The use of berberine is promising, and the manuscript addresses a knowledge gap in topical treatment of viral conditions. The rationale of choice of excipients is also a point of strength in the manuscript. References are adequate and up to date. I have some minor comments for the authors before the manuscript can be accepted for publication:

1- Please add a small paragraph at the end of the introduction section to recapitulate the aim of work, and to highlight the novelty of the manuscript.

2- The authors need to comment on the stability of berberine in the formulation, since it is both heat and light sensitive.

3- Section 2.5, if the UPLC method was adopted from a reference, it needs to be stated

4- Figure 7 the S.D. values are very high, it is always advised that if the ex vivo permeation studies shows variability when it is conducted in triplicate, it is best to conduct it 6 times instead (n=6)

5- The conclusions section is very lengthy, please only summarize the main findings

Author Response

  • Reviewer: Please add a small paragraph at the end of the introduction section to recapitulate the aim of work, and to highlight the novelty of the manuscript.
  • Comment: We added the relevant paragraph and modified the section.
  • Reviewer:The authors need to comment on the stability of berberine in the formulation, since it is both heat and light sensitive.
  • Comment: We added the relevant information in the annex.
  • Reviewer:Section 2.5, if the UPLC method was adopted from a reference, it needs to be stated
  • Comment:We added the relevant paragraph.
  • Reviewer:Figure 7 the S.D. values are very high, it is always advised that if the ex vivo permeation studies shows variability when it is conducted in triplicate, it is best to conduct it 6 times instead (n=6)
  • Comment:We run the experiments to increase the numerosity
  • Reviewer:The conclusions section is very lengthy, please only summarize the main findings.
  • Comment: We rewrote the section.